# Magnetostrictive Guided Wave Technique Verification for Detection and Monitoring Defects in the Pipe Weld

**DOI:** 10.3390/ma12060867

**Published:** 2019-03-15

**Authors:** Se-Beom Oh, Yong-Moo Cheong, Deok-Hyun Lee, Kyung-Mo Kim

**Affiliations:** 1Nuclear Materials Research Division, Korea Atomic Energy Research Institute, 989-111, Daedeok-daero, Yuseong-gu, Daejeon 34057, Korea; ymcheong@kaeri.re.kr (Y.-M.C.); dhlee1@kaeri.re.kr (D.-H.L.); kmkim@kaeri.re.kr (K.-M.K.); 2Department of Materials Science and Engineering, Dankook University, Dandae-ro, Dongnam-gu, Cheonan 31116, Korea

**Keywords:** long range inspection, guided wave, magnetostrictive sensor, signal processing

## Abstract

During inspection of piping in nuclear power plants or other industries, it is difficult to implement conventional nondestructive techniques due to limited accessibility or obstacles such as pipes with insulation, pipes buried underground, structural complexity, or radiation environments. In addition, since the defects mainly occur in the weld region or support area, it is not easy to separate defect signals from those of structural components. To solve these problems, we developed a technique to detect and monitor the formation and growth of defects, using a magnetostrictive guided wave sensor. This sensor has advantages (such as sharp and clear signal patterns and ability to easily eliminate the signal from the geometric structure) over the conventional piezoelectric transducer. To verify our technique, signals from actual pipe welds with defects were acquired and processed with our phase matching/subtraction program. The proposed technique shows a superior capability for detection and monitoring of defects, compared to the conventional guided wave methods.

## 1. Introduction

Nuclear power plants have found various defects that were not predicted at the time of design. To cope with this, various advanced nondestructive evaluation technologies related to the safety of nuclear power plants are being developed [1]. A long-range guided wave ultrasonic has been introduced as a non-destructive testing method that diagnoses defects in buried pipes or claddings that are difficult to test with current technology [2,3]. Guided wave ultrasonic methods are being used for large inspection areas in the case of thin plates or pipes, measuring from a few meters or tens of meters from where the sensor is located. The accuracy is lower than that of local inspection, but it can be efficiently performed for long distance inspection. It can also be used in cases where it is difficult to access specimens that are covered or buried underground [4]. Choosing the wanted wave mode from among the many theoretically predicted modes is an important consideration. In consideration of dispersion characteristics, scattering characteristics, and ultrasonic generation method of each mode, it is necessary to select a mode with characteristics most suitable for the specimen. The mode of the guided ultrasonic wave thus selected can be determined by analyzing the reflected signal from the defects in the propagation path. Because the guided ultrasonic waves propagate along the medium, the defect position is determined by signals reflected from points such as notches or corrosion defects that cause changes in the pipe’s cross-sectional area [5,6].

However, the actual ultrasonic signal is complicated due to the geometrical complexity of the welded portion, the heterogeneous connection part, the deformed pipe, the pipe support, and the various valves. Also, most of the defects’ signals are generated at welds and pipe supports, making it difficult to distinguish between them [7]. It is difficult for guided wave ultrasonic testing of long-distance piping to distinguish between internal/external defects. Therefore, it is necessary to develop non-destructive testing (NDT) methods to assess the reliability of pipe structures and components [8,9]. A guided wave ultrasonic signal that uses a magnetostrictive sensor can generate an almost theoretical waveform structure because the signal waveform is cleaner than other piezoelectric ultrasonic sensors or electromagnetic acoustic sensors. In addition, it can be permanently installed and continuously monitored [10]. One way to simplify the detection of damage is to compare a received signal during the operational life of the structure with an undamaged signal (reference signal). The simplest way to do this is to subtract the two signals. Because the reflections from structural features remain unchanged, they are removed on subtraction. This means that it is no longer critical to separate feature-reflections from defect-reflections in the time domain [11,12].

With these advantages, we developed a structural health monitoring (SHM) technique that uses magnetostrictive guided wave methods. A magnetostrictive guided wave sensor was permanently installed in the pipeline, where it periodically collected and compared signals to determine defect generation and growth. For a more accurate distinction, a vibration mode with low dispersibility should be selected. The dispersibility of the transmitted ultrasonic waves should be low so that the noise signal can be minimized. It is advantageous to use a torsional vibration T (0, 1) mode with no radial and axial displacement vectors and only circumferential displacement at the acoustic attenuation on the side to detect and monitor micro-defects in long-range pipes [13]. Temperature perturbations affect the ultrasonic velocity in the specimen and cause a fine variation in the measured signal in time domain. Temperature perturbations lead to changes in the geometrical and physical properties of the structure, causing significant changes in the measured ultrasonic signals. This is not desirable, as the SHM system may misinterpret the changes as being caused by damage, when this is not the case. Some researchers have sought a single parameter that offers an indication of damage but is independent of environmental conditions, whereas others have used a knowledge-based approach with positive results. Therefore, it is necessary to correct the error caused by temperature or other variables, depending on the ambient environment of the specimen. [14,15].

In this study, a magnetostrictive sensor was permanently installed in the piping, and the guided wave ultrasonic signals were periodically collected and compared with the reference signal. This made the initial crack detection and defect growth monitoring possible by observing signal changes due to variations in the cross-sectional areas of the piping. A signal phase matching technique was also developed to precisely compare the temperature changes in the guided ultrasonic signal and the waveform variation due to environmental variables. We tested the welded pipe for notched and corrosion defects and verified the developed technology.

## 2. Long-Range Guided Wave Ultrasonic Dispersion

Guided ultrasonic waves are a type of elastic wave propagating along the geometrical assembly of a structure, caused by superimposition and interference of an initial excitation wave. Therefore, guided waves have different characteristics from ordinary bulk waves. They have a dispersion characteristic in which the propagation speed varies, depending on the frequency, acoustic characteristics of the material to be inspected, structure, and specification [16].

Due to the dispersion characteristics of the guided ultrasonic waves, it is necessary to create a dispersion diagram based on the characteristics of the object, and select a vibration mode and frequency parameter suitable for the purpose. It is important to find a mode that offers suitable characteristics (such as dispersion and ultrasonic wave excitation) for non-destructive inspection of a material [17]. It is advantageous to detect defects by selecting a vibration mode and frequency with small variation in phase velocity and group velocity based on frequency change [18]. We designed and manufactured a magnetostrictive sensor capable of torsional vibration T (0, 1), wherein the group velocity is frequency independent and nondispersive. Figure 1 shows the dispersion diagram of the 60-mm diameter carbon steel pipe used in this study.

## 3. Magnetostriction Phenomenon and Application

Magnetostriction is a phenomenon in which mechanical deformation occurs when a ferromagnetic material is placed under a magnetic field; it was discovered by Joule in 1847 [19]. The reverse phenomenon is when the magnetic state inside the material changes when there is stress on the material. This is called the inverse magnetostrictive effect, discovered by Villari in 1864. Magnetostriction that is defined by partial length change when the ferromagnetic material is placed under a magnetic field is represented by Equation (1):(1)$$\lambda =\frac{\Delta l}{l}$$

In this equation, *λ* is the magnetostriction, and *l* and ∆*l* is the length before deformation and length changes due to deformation, respectively. Magnetostriction is divided into spontaneous magnetostriction and magnetostriction by the addition of an external magnetic field. Spontaneous magnetostriction is a term used when a ferromagnetic material passes through the Curie temperature, forming a magnetic domain. Magnetostriction by an external magnetic field is a term used when a magnetic domain modifies rearrangement owing to an external magnetic field that is applied in the material, generating a magnetic domain. This is explained in Figure 2: (a) shows the atomic arrangement of the ferromagnetic material at a temperature higher than the Curie temperature. When the temperature is lowered, magnetostriction occurs as shown in (b) by spontaneous magnetization. When a magnetic field is applied externally, the magnetic moment of the atoms in the magnetic domain is aligned in one direction, as shown in (c), and magnetostriction occurs. The non-destructive magnetostrictive sensor utilizes the magnetostrictive property of the external magnetic field. That is, the magnetostriction generated when changing the intensity of the external magnetic field is used as the source of the ultrasonic wave energy [20,21,22].

## 4. The Guided Ultrasonic Wave Signal Phase Matching and Subtraction

The guided wave ultrasonic signals received from the pipeline are shown in Figure 3a. The indicated defects, such as those in the welds, piping supports, and heterogeneous connections, were due to deposits existing on the inside/outside of the geometric structure. Most of the defects occurred at the same position as the geometric signals. Signal size was also relatively small, which made defect detection difficult [23]. In order to distinguish a small signal, the defect signal can be extracted by removing the geometric signal from the reference signal [24,25,26]. Figure 3b is a graph obtained by simply subtracting a waveform of the reference signal from a waveform of the measurement signal. However, it is hard to simply detect defects because arithmetically removed signals are difficult to extract from geometric signals. The measurement signal changes due to fine sound velocity change and zero point movement, even if the same probe and measurement parameters are used at the time of measurement [27]. We developed an algorithm and a program that adjusts the guided ultrasonic signal to remove the reference signal in order to accurately extract only the defect signal. The reference signal for ultrasonic signal monitoring was divided into the difference between the zero point (which varies with the measurement), and the proportional change due to fluctuations in sound velocity based on the temperature.

The time function of the reference signal is F(t) and the time function of the measurement signal is F′(t). The measurement signal is shown in Figure 4. The difference between amount A and proportional change B due to the zero movement must be corrected. The induction ultrasound phase matching/subtraction algorithm was developed to determine two unstable structural signals *t*_1_, *t*_2_ at the same position, such as the welded part signal, to match the two signals and then to eliminate them. After moving the entire signal by the zero point difference $t_{1}{}^{\prime }-t_{1}$ to match one peak, the remaining signals are matched from $t_{2}{}^{\prime }$ to $t_{2}$ using a coefficient $X\;\left(\frac{t_{2}-t_{1}}{t_{2}{}^{\prime }-t_{1}}\right)$, which reduces the signal in the range to a constant ratio. The equation for matching the measured signal F′(t) with the reference signal is as follows:(2)$$F\left(t\right)=\left(t_{1}{}^{\prime }-t_{1}\right)+F^{\prime }\left(t\right)\cdot \left(\frac{t_{2}-t_{1}}{t_{2}{}^{\prime }-t_{1}}\right)$$

The difference in the time axis can be regulated into a proportional change based on the sound velocity difference and a change in the absolute value based on the change in the zero point, so that the known geometrical signals can be normalized and removed.

## 5. Experimental Verification of Defects in Welded Pipes

### 5.1. System Set-Up

The dimensions of the pipe used for the experiment were length of 3 m and diameter of 60 mm. The material used was low carbon steel SA 106 Gr. B. An overview of the system’s set-up is shown in Figure 5. In order to investigate the effect of the weld, two pieces of pipe were welded. Test frequency was 64 kHz, and the torsional vibration T (0, 1) mode was used, which has no dispersion in the frequency band. The welds were done 2 m from the end of the pipe to distinguish additional signals. A magnetostrictive ultrasonic sensor was installed at a distance of 0.3 m from the end of the pipe to differentiate between the unpredictable signals that are reflected. Therefore, the signal of the weld is measured at a distance of about 1.7 m and the back wall signal is measured at about 2.7 m. All signals were received by MsS 2020 guided ultrasonic equipment (SwRI, San Antonio, TX, USA). The signal processing was a simple subtraction done through the Inspection program of the MsS, and phase matching and subtraction was performed using the developed program.

### 5.2. Generation/Propagation of Torsional Wave Vibration Mode

The torsional wave vibration T (0, 1) mode is an axisymmetric mode; thus, the guided wave ultrasonic is propagated in the axial direction. A ferromagnetic Fe-Co-V strip with excellent magnetostriction properties is attached in the circumferential direction of the pipe using an epoxy adhesive. The permanent magnet is rotated in the circumferential direction of the magnetostrictive strip so that the direct current bias magnetization is induced in the strip. A coil is wound on the strip and an alternating current is applied, forming a magnetic field in the axial direction. Shear displacement occurs in the strip due to interaction with the DC (Direct current) bias magnetization generated in the circumferential direction. The shear displacement generated in the magnetostrictive strip is transmitted to the pipe while passing through the epoxy, which is the contact medium, and the torsional vibration T (0, 1) mode propagates along the pipe [28].

### 5.3. Notch and Corrosion Defect Verification at the Welded Pipe

The notch was machined perpendicular to the axis of the pipe. The width of the notch was about 1 mm, and depth was 5-30% of the thickness of the pipe. In addition, the reduction of the cross-sectional area, based on the depth of the notch, was observed. The defect signal was determined to be the remaining signal that was removed by simple subtraction and phase-matching subtraction from the reference signal.

Table 1 shows the notch depth converted to the cross-sectional area (CSA) ratio. Figure 6 is the result of simple subtraction of the reference signal according to the depth of the notch defect size. Because the position of the detected signal coincided with the actual position of the notch, it was possible to identify the signal by the notch. The notch signal was created when the cross-sectional area loss rate reached about 1% or more. The size of the defect signal after the subtraction changed proportionally, when the cross-sectional area’s loss was more than the point at which the defect signal is distinguished. It was determined that the cross-sectional loss rate can be predicted by using the size of the removed signal in the case of the notch defect on the pipe. Figure 7 is a comparison of the signal removed by the phase matching program and the simple elimination signal. The results of this experiment show that the noise signal was reduced when compared with the simple subtraction signal, and the pseudo signal (not the actual notch signal at the arrow position) was almost completely eliminated. It was confirmed that when the reference signal was removed by applying the program to the area in the pipe where the notch was formed in the welded portion, only the accurate notch signal was left.

The piping conditions were the same as those used for notch defect verification; the conditions of the frequency and vibration modes were also consistent. Pipe holders were made, and 30% nital solution was applied to the pipe welds. Corrosion signals were checked every 24 h. In order to prevent the corrosive products generated during the corrosion test process from affecting the surface corrosion, the signal was measured after removing the holder. The decrease in the pipe’s cross-sectional corroded area was measured, and the change in signal size was confirmed. The signal in each condition was compared (using the proposed program) with the signals extracted by simple subtraction and phase matching subtraction. Table 2 shows the ratio of the pipe’s cross-sectional area loss. The surface condition of the corrosion defect is shown in Figure 8. The length of the defect along the pipe’s axis is denoted as A to B and measured to be approximately 0.12 m. Figure 9 compares the simple elimination of the signal with the phase matching program at a cross-sectional loss of 1.66%.

A corrosion defect signal was detected at a cross sectional loss rate of 1.66% and above. Figure 9b shows that the amplitude of the defect signal increased by about 0.2 V by using the phase matching program, when compared to the simple elimination signal. Because the amplitude of the defect signal from which the reference signal was removed grew relatively, the corrosion signal was clearly distinguished. Figure 10a,b show the simple elimination results of the reference signal when the cross-sectional area loss rates were 2.55% and 3.46%, respectively. The larger the cross-sectional loss area, the larger the size of the simple elimination signal. However, although the signal was simply removed, the surrounding pseudo signal still appeared.

Figure 11 is a comparison of the signal extracted from the phase matching-subtraction program and the simple elimination method. The diagrams in (a) and (b) show the subtracted signals, and the figure below shows the result of using the program. The distance between the two peaks was about 0.12 m in the phase matching subtraction program (Figure 11); this coincided with the distance between A and B where the actual corrosion occurred (Figure 8). With this, the corrosion defect length can be predicted using the signal from which the reference signal was removed. It was difficult to completely eliminate the pseudo signals and noise appearing throughout the specimen, but the error rate was minimized at the point where the corrosion signal appeared. The magnitude of the detected corrosion signal increased by 1.5 to 2 times (about 0.1 V increase) compared to the previous method. In addition, the detection performance of the corrosion defect signal for the ambient noise improved.

## 6. Discussion and Conclusions

In order to monitor defect generation and growth in pipes with limited accessibility (such as insulation or piping), a magnetostrictive guided wave sensor was permanently installed in the piping to enable periodic monitoring and subsequent signal collection.

We developed a phase matching/subtraction algorithm and program to detect defects and monitor growth. We also eliminated the reference signal from the collected signal to improve defect detection ability by removing most structural signals. As a result, we were able to predict the size and length of the defect.

The signals that identified defect generation and growth were collected, processed, and evaluated in actual welded piping to verify the feasibility of the technology. The defect detection capability (~1% CSA) was found to be superior to the defect size of 5~10% CSA, which is the detection limit of conventional guided ultrasonic technology. 

Our technique, however, has limitations. It seeks to calibrate the entire signal based on the peaks of the start point and end back wall signals. In the process, a slight error occurs between the defective signals located in the middle. A better result can be obtained if phase matching of multiple peaks is achieved. We believe that this method can be applied in future to insulation material pipes or buried pipes.

## Figures and Tables

**Figure 1 materials-12-00867-f001:**
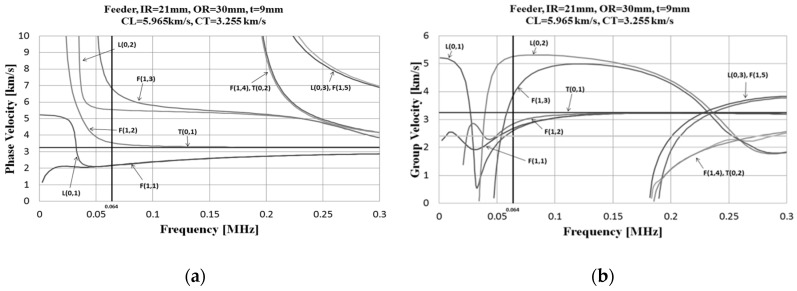
Curves of the (**a**) phase velocity and (**b**) group velocity of a carbon steel pipe with inner radius of 21 mm and outer radius of 30 mm.

**Figure 2 materials-12-00867-f002:**
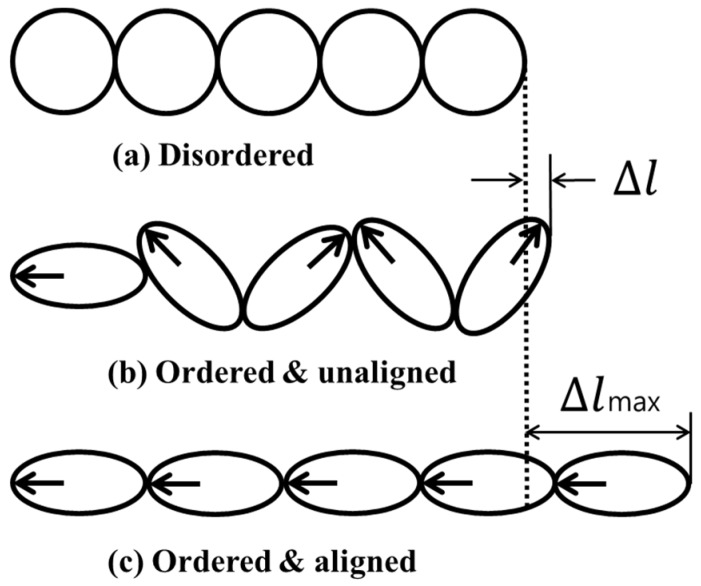
Magnetostriction: (**a**) disordered, (**b**) ordered & unaligned, (**c**) ordered & aligned.

**Figure 3 materials-12-00867-f003:**
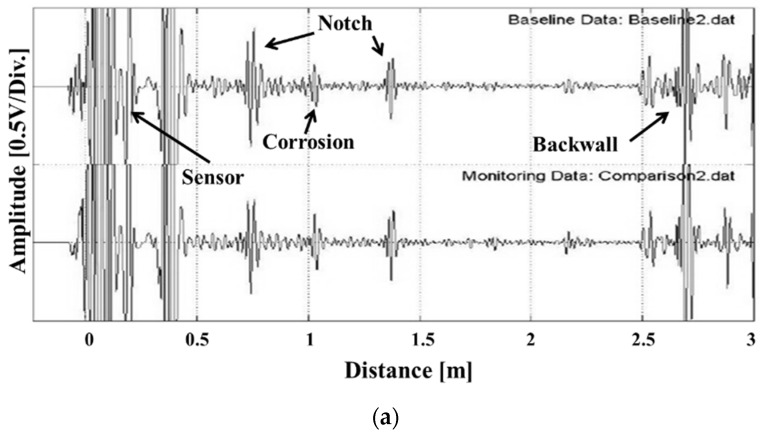
(**a**) Typical guided wave signal acquired from a pipe by a magnetostrictive guided wave transducer and (**b**) simple subtraction of guided waves from the reference signals show residual components due to differences in the experimental conditions.

**Figure 4 materials-12-00867-f004:**
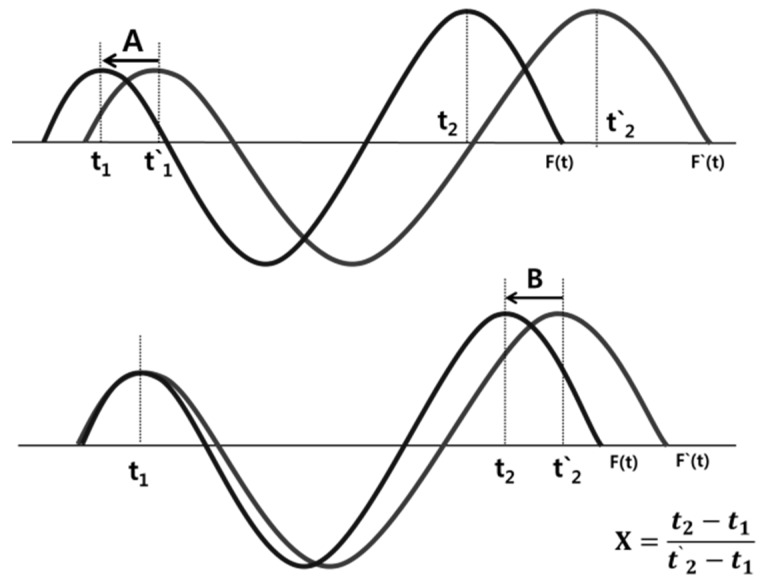
Schematic diagram of a phase matching algorithm between the reference signal and measurement signal.

**Figure 5 materials-12-00867-f005:**
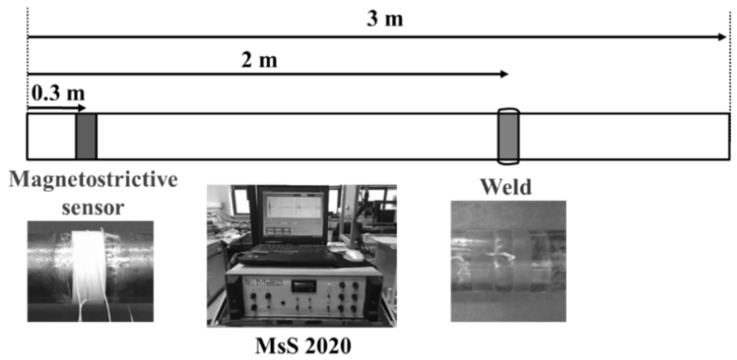
The schematic diagram of the magnetostractive wave guide ultrasonic monitoring system.

**Figure 6 materials-12-00867-f006:**
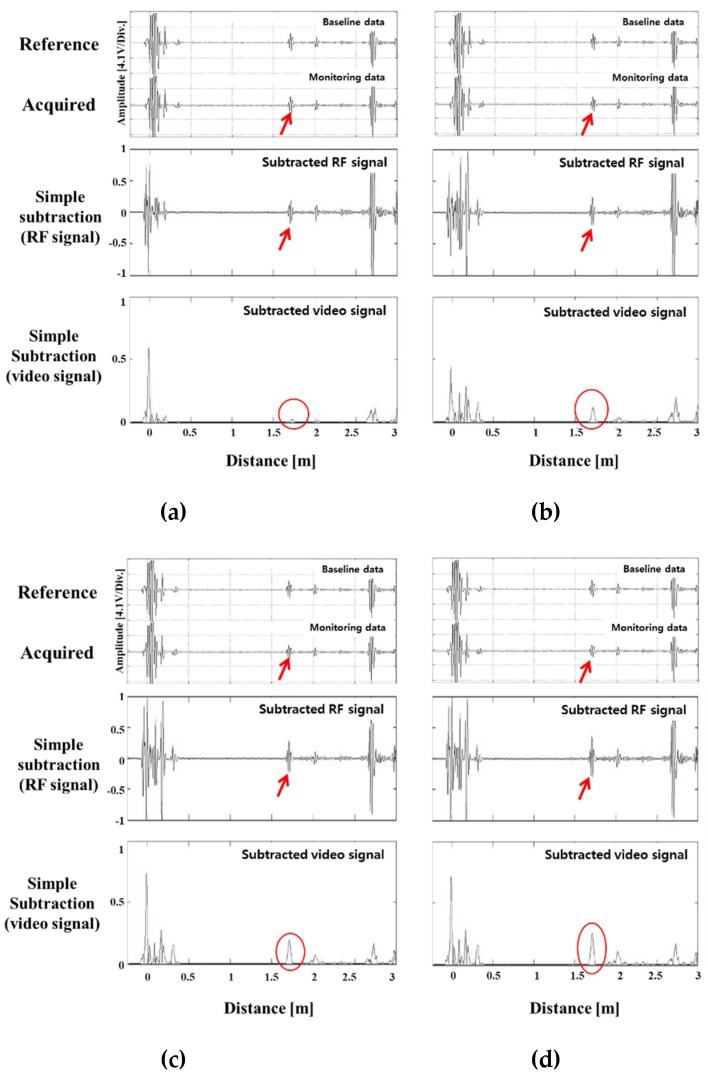
Notches with various depths in a pipe weld: (**a**) 5%, (**b**) 10%, (**c**) 20% and (**d**) 30% of pipe wall thickness. The reference signal (top), acquired signals (middle), and signals after simple subtraction (bottom).

**Figure 7 materials-12-00867-f007:**
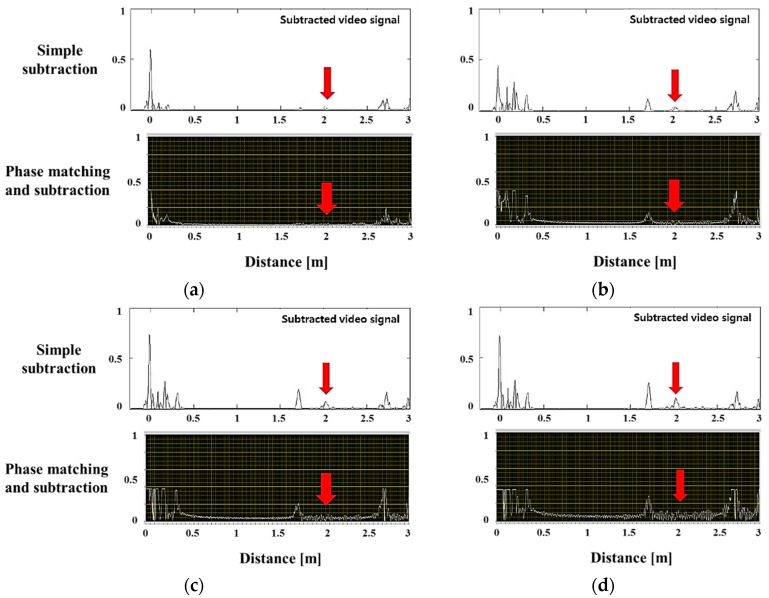
Subtraction between simple elimination, and phase matching-subtraction; signals of notches with various depths in a pipe weld: (**a**) 5%, (**b**) 10%, (**c**) 20% and (**d**) 30% of pipe wall thickness.

**Figure 8 materials-12-00867-f008:**
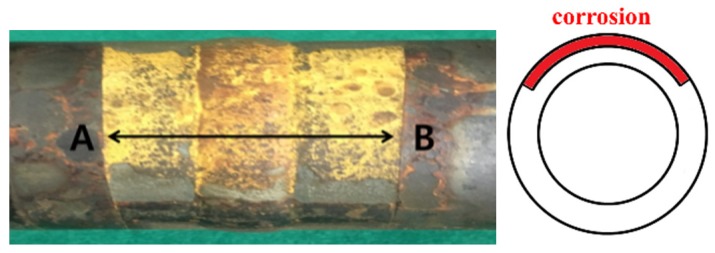
Pipe surface after corrosion (**left**) and diagram of cross-sectional area loss (**right**).

**Figure 9 materials-12-00867-f009:**
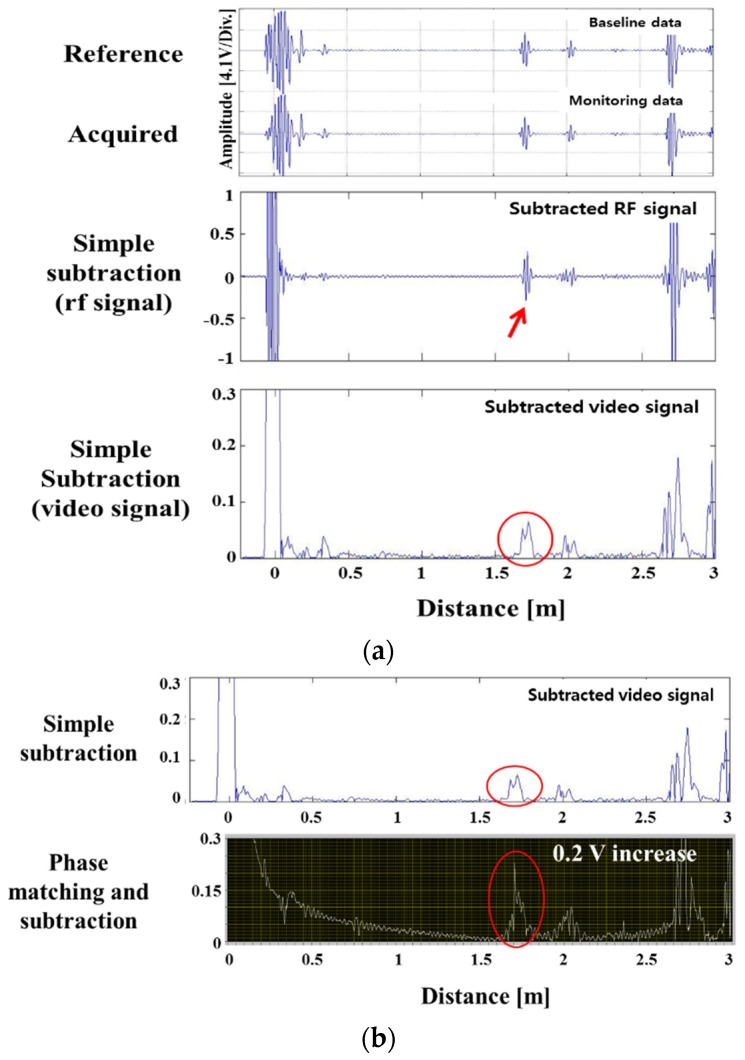
(**a**) Defect signals of surface corrosion in a pipe weld with depths, (**b**) comparison of subtraction between simple elimination and phase matching-subtraction; signals of corrosion in a pipe weld (defect ratio: 1.66%).

**Figure 10 materials-12-00867-f010:**
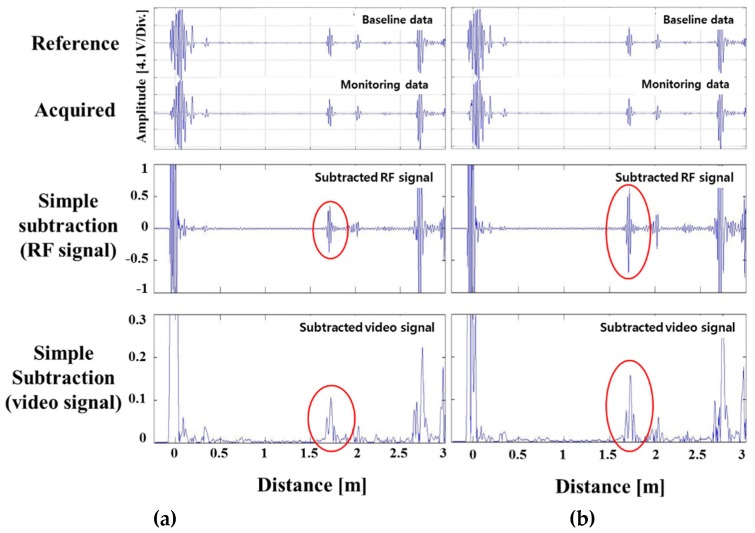
Signals of surface corrosion with various depths in a pipe weld: (**a**): defect ratio 2.55%, (**b**): defect ratio 3.46%.

**Figure 11 materials-12-00867-f011:**
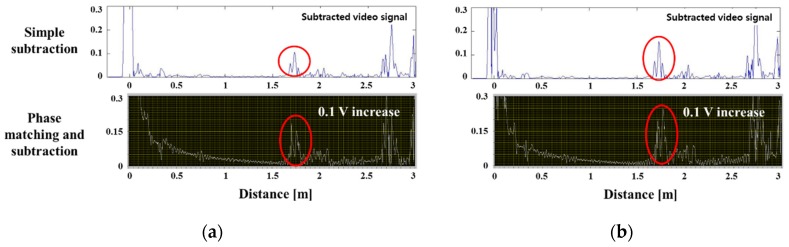
Comparison between simple elimination and phase matching-subtraction; signals of corrosion in a pipe weld: (**a**) defect ratio 2.55%, (**b**) defect ratio 3.46%.

**Table 1 materials-12-00867-t001:** Notch depth to pipe wall thickness and notch area to the CSA of pipe at the weld region.

Notch Depth (%)	Defect Ratio (CSA %)
5	0.48
10	1.0
20	2.8
30	4.9

**Table 2 materials-12-00867-t002:** Corrosion depths to pipe wall thickness and corrosion area to the CSA of pipe in the weld region.

Corrosion Time [hr]	Defect Ratio (CSA %)
24	1.66
48	2.55
72	3.46

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
