# Peer review of "Magnetostrictive Guided Wave Technique Verification for Detection and Monitoring Defects in the Pipe Weld"

_materials, 2019, doi:10.3390/ma12060867_

Reviewer 1 Report

 Experiments with advanced baseline subtraction method (BSA) were presented in a paper.

It is recommended to show a sketch of a tested pipe (or a photo) with probe position in the reference to the pipe ends.

It is recommended to make Figures 5 -10 larger in size and in font. Right now it is very hard to see signals and what is accomplished.

Based on reflection pattern, it looks like a probe was installed at a distance about 30 cm from the end of the pipe. This is why a pipe end reflection is observed at a distance 2.7 meters.

Due to some reasons, the results of subtraction still have high residual amplitudes where the pipe end is located. For example on Figure 5, these residual remains have amplitudes equivalent to the 10 - 15% anomaly. This residual signal amplitude is not getting any smaller when advanced subtraction algorithm is applied.

For laboratory experiment, the conventional baseline subtraction works very well. Example of utilizing conventional BSA during experiments with hot pipes using magnetostrictive sensors can be found in these referenced papers: 

 Vinogradov S, Eason T, Lozev M, ‘Evaluation of Magnetostrictive Transducers for Guided Wave Monitoring of Pressurized Pipe at 200ºC,’ in Journal of Pressure Vessel Technology, Vol. 140, April 2018, pp 021603-1.

Vinogradov S; Fisher J, Review of Guided Wave Testing Using Magnetostrictive Transducers. In Proceedings of the 12th European Conference on Non-Destructive Testing, Gothenburg, Sweden, 11–15 June 2018.

It is recommended to provide a clear description of experimental setup including type of ferromagnetic patch material, method of bonding, temperature records and software/algorithm used for data analysis. 

It should be also clearly shown that this advanced algorithm can provide a higher accuracy not only for a local region of a pipe (welded area) but also for the entire length of the pipe. If this can not be accomplished then the reasons should be explained.

Author Response

 Thank you for your interest in this paper and your good comments and answers. Problems related to the resolution of the picture have been revised. As you mentioned, I have attached an overview of the experiment to enhance understanding.

 As the reflection (diffused reflections) are made a lot with the air with the low impedance at the end of the pipe, the signal which does not have a certain pattern is difficult to interpret. In this study, we focused on the evaluation of the signals generated in the welds. Because dangerous defects are mainly generated in welds and are difficult to observe. If I can design a more precise method and analyze the signal at the end back wall signal, I think it will be possible to increase the resolution of defects that may be generated at the whole of the pipe.

 Reviewer 2 Report

In this manuscript, the authors developed a technique to monitor the defects in pips by combing magnetostrictive sensor with a phase matching and subtraction algorithm program. Their results show that :1. comparing to guided ultrasonic waves, the torsional vibration T(0,1) mode of the magnetostrictive sensor has a constant group and phase velocity. 2. the phase matching and subtraction program produces weaker noise signal and stronger defect signal comparing to the simple subtraction method. Overall, I think this work is interesting and important. There are a few questions/comments that I would like the authors to address in a revised manuscript.

1. As the authors mentioned in section 3, ferromagnetic materials become paramagnetic at Curie temperature, thus exhibiting a large magnetostriction. However, the Curie temperature for most materials are extremely high. For example, it is for 1043K for Iron. In this work, the authors used a “Fe-Co-V strip with excellent magnetostriction properties”. How was the degree of magnetostriction determined for this material? Are all the experiments/data collection done under room temperature? 

2. Line 112, how was the reference signal generated and collected?

3. Line 160 and line 191, the authors referred to table 1 and 2 but these tables are missing in the manuscript.

3.For section 5.1 and 5.2, it would be better to draw an illustration of the setup, label the pipe, magnetostrictive strip, magnet, the position of the notch and weld, etc.

4. The resolution of figures 5-6, 8-10 are low so it is hard to see the signals clearly. These figures also need to be enlarged.

5. The title of section 6 should be “Discussion and Conclusion” since the “Results” have been presented in section 5.

6. Line 236, the authors claim that they “were able to predict the size and length of the defect”. However, their data only show that for a larger defect, the defect signal is stronger. The “size and length” information can not be obtained. Can the authors build a mathematical relationship between the intensity of signal and the size/ratio of defect?

7. The phase matching and subtraction method can almost eliminate the pseudo signal for the notch defect while does not improve the actual notch signal, however, it increases the corrosion signal but does not suppress the pseudo signal for the corrosion defect. Can the authors offer an explanation why this method shows dramatic different effects on different defects?

Author Response

Thank you for your interest in this paper and your good comments and answers. I have made corrections depending on your comments.

Here are answers to some questions.

1. As the authors mentioned in section 3, ferromagnetic materials become paramagnetic at Curie temperature, thus exhibiting a large magnetostriction. However, the Curie temperature for most materials are extremely high. For example, it is for 1043K for Iron. In this work, the authors used a “Fe-Co-V strip with excellent magnetostriction properties”. How was the degree of magnetostriction determined for this material? Are all the experiments/data collection done under room temperature?

 -> Because the experimental temperature was at room temperature, we could not create a curie temperature environment. Therefore, magnetic field is generated in the domain by moving the ferromagnetic magnet in one direction to the strip. As a result of the previous experiment, it was confirmed through experiments that the generated magnetism is stably maintained in the period of one year or more.

2. Line 112, how was the reference signal generated and collected?

->This signal is usually a signal obtained by measuring a same pipe using a magnetostrictive sensor. The pipe support, corrosion/notch defects were generated and measured in the same size piping to verified the typical signal shape. I dropped each one so that do not signal overlap occurred.

6. Line 236, the authors claim that they “were able to predict the size and length of the defect”. However, their data only show that for a larger defect, the defect signal is stronger. The “size and length” information can not be obtained. Can the authors build a mathematical relationship between the intensity of signal and the size/ratio of defect?

->Mathematical construction is hard, but the width of the crack can be determined by directly measuring the distance between the peaks of the signal. The length of the crack seems to be wrong because it determines the degree of defect relative to the cross-sectional area of the pipe. The length of the crack can not be known, but the approximate crack size can be predicted and the width of the corrosion defect can be predicted.

7. The phase matching and subtraction method can almost eliminate the pseudo signal for the notch defect while does not improve the actual notch signal, however, it increases the corrosion signal but does not suppress the pseudo signal for the corrosion defect. Can the authors offer an explanation why this method shows dramatic different effects on different defects?

->Corrosion defects form over a wide range and do not occur at a constant depth on the surface. Therefore, the cross-sectional area in the vertical direction is not constant in each pipe because the defect is criminated by cross sectional area loss of the pipe, the received signal is not like the notch defect. Due to the limitations and errors of the measuring equipment averaging 100 times per second, it was difficult to completely remove the pseudo signal. However, the typical corrosion defects signal size is very small and difficult to measure. Therefore, we wanted to maximize the effect of increasing the signal size and to improve the defect detection capability.

 I hope this is the answer to your question.

Thank you

Yours sincerely

 Reviewer 3 Report

Dear Authors,

The paper seems to be valuable, but before publication, I recommended a few important issues for consideration. In my opinion, the paper can be published, however….

Please strongly underline what is new (!) and what kind of development we can see in your paper. I suggested a major revision. Please – check carefully the English part (and style edition as well) of the submitted manuscript.

1.       The introduction is too short. A valuable will be to extend this section using more references. Please, present “state of the art” in this section brief discussion of the placement of this paper. In literature, exist a lot of papers devoted to this topic.

2.       Line 72 – what kind of material was used for the pipe? Steel? Please provide as much as possible details about this material; chemical composition, microstructure, basic mechanical properties etc.

3.       Line 80 – please describe this effect in details. Also in literature is well known inverse effect – see Villari effect. It seems that this point is crucial for readers while understanding the idea of the proposed method

4.       Eq 1 – It seems that lambda is an ordinary strain – please extend the part of magnetostriction and define it again. Show connections between mechanical and magnetic signals.

5.       Fig. 3a – Please scale y-axis (!) and try to improve the quality of the remaining diagrams

6.       Please briefly discuss the advantages and disadvantages of the presented approach. For instance how about non-ferromagnetic materials?

7.       It will be good to discuss the measurements errors and limitations of the proposed technique

8.       Please add real measurement stand with the description of each device used for the non-damage test.

Author Response

Thank you for your interest in this research paper. I have completed the modifications according to your comments.

6.  Please briefly discuss the advantages and disadvantages of the presented approach. For instance how about non-ferromagnetic materials?

-> We used Ni ferromagnetic strips, which are generally used, to test them. However, it was difficult to distinguish the received signal at our system because of the high noise level. The Fe-Co-V strip is relatively good in terms of signal to noise ratio and that is advantage to comparing signals.

7. It will be good to discuss the measurements errors and limitations of the proposed technique

->In this study, most of the values are relative to the reference signal because of the signals are compared and subtracted. Therefore, it is difficult to calculate the error, but if it is converted from the time domain, an error of about 0.05 us.

This technique also had some limitations. The current method is to calibrate the entire signal based on the peak of start point and peak of end back wall signal. In this case, a slight error occurs between the defective signals located in the middle. If the phase matching of multiple peaks can be achieved, a better result can be obtained.

I hope this is the answer to your question.

Thank you

Yours sincerely

 Round  2

Reviewer 1 Report

In abstract the statement "The technique can be applied to the examination of insulated pipe or pipe buried underground" is not supported by experimental data. If it is an assumption it is better to use it in the paper body.

In the sentence "A guided wave ultrasonic signal that uses the 52 magnetostrictive sensor can generate an almost theoretical waveform structure because the signal waveform 53 is cleaner than other piezoelectric ultrasonic sensors or electromagnetic acoustic sensors. Also, it can be 54 permanently installed and monitored continuously [10], the reference 10 is relevant to piezo transducers. I would suggest to find a reference to magnetostrictive transducers.

Figure 3 is missing a and b marks on plots.

Table 1 is not necessary. It would be enough to say that the mockup is made of  low carbon steel material.

Author Response

Thank you for your good feedback.

I deleted the sentence mentioned in the abstract according to your comments.("The technique can be applied to the examination of insulated pipe or pipe buried underground")

I agree with the fact that the contents of Reference 10 are slightly different from the paper, and found a new relevant reference. Figure 3 was modified, and Table 1 was deleted and referred to "low carbon steel" in the paper.

Thank you

Yours sincerely,

Reviewer 2 Report

Overall, the revised manuscript is improved with the newly added figure, tables, discussion and references. There is only one question/concern left I would like the authors to answer in a further revised manuscript. The authors claim that “the approximate crack size” and “width of the corrosion defect can be predicted”. However, how the prediction works based on their measured data is unclear in their manuscript. They need to show at least one example of predicting the size of crack and width of corrosion from their data.

Author Response

Thank you for your interest and advice. I modified the contents of corrosion length in the revision manuscript.

Thank you for helping to improve the quality of this manuscript.

Reviewer 3 Report

Dear authors,

I appreciate the effort to improve the manuscript. The paper now looks better. I am in agreement with it with a minor correction. Namely: Table 1 shows the mechanical properties - please specify the source (if it is not own research) and specify the mechanical quantities in the SI system (e.g. MPa etc.).

Good luck

reviewer

Author Response

Thank you for helping to improve the quality of this manuscript. The piping used in the study is a piping made according to the ASME standard, I added the contents to the reference.

Yours sincerely,